# Health Information Technology and Doctor Shopping: A Systematic Review

**DOI:** 10.3390/healthcare8030306

**Published:** 2020-08-28

**Authors:** Clemens Scott Kruse, Brady Kindred, Shaneel Brar, Guillermo Gutierrez, Kaleigh Cormier

**Affiliations:** School of Health Administration, Texas State University, San Marcos, TX 78666, USA; b_k118@txstate.edu (B.K.); skb133@txstate.edu (S.B.); g_g251@txstate.edu (G.G.); kpc16@txstate.edu (K.C.)

**Keywords:** prescription drug monitoring program, pharma cloud, doctor shopping, PDMP

## Abstract

Doctor shopping is the practice of visiting multiple physicians to obtain multiple prescriptions. Health information technology (HIT) allows healthcare providers and patients to leverage records or shared information to improve effective care. Our research objective was to determine how HIT is being leveraged to control for doctor shopping. We analyzed articles that covered a 10-year time period from four databases and reported using preferred reporting items for systematic reviews and meta-analysis (PRISMA). We compared intervention, study design, and bias, in addition to showing intervention interactions with facilitators, barriers, and medical outcomes. From 42 articles published from six countries, we identified seven interventions, five facilitator themes with two individual observations, three barrier themes with six individual observations, and two medical outcome themes with four individual observations. Multiple HIT mechanisms exist to control for doctor shopping. Some are associated with a decrease in overdose mortality, but access is not universal or compulsory, and data sharing is sporadic. Because shoppers travel hundreds of miles in pursuit of prescription drugs, data sharing should be an imperative. Research supports leveraging HIT to control doctor shopping, yet without robust data sharing agreements, the efforts of the system are limited to the efforts of the entity with the least number of barriers to their goal. Shoppers will seek out and exploit that organization that does not require participation or checking of prescription drug monitoring programs (PDMP), and the research shows that they will drive great distances to exploit this weakest link.

## 1. Introduction

### 1.1. Rationale

The World Health Organization (WHO) encourages providers to remove impediments that prevent a patient’s voice from being heard [1,2,3]. However, the recent opioid epidemic serves as a reminder of systemic inefficiencies that push patients’ voice into unhealthy practices. Unchecked, these inefficiencies are deadly [4]. An example of this can be described by the practice of doctor shopping, which is the practice of individuals visiting several physicians to obtain multiple prescriptions or to attain a preferred medical diagnosis without distinct material gain [4]. It is also defined by imposing a threshold of 6 or more prescriptions from at least 6 different prescribers within 6 months’ time [4]. Such a specific description made our team wonder why health information technology (HIT) could not be leveraged to control for this practice. The simple answer may lie in the concept of interoperability or lack thereof because even countries with a national health system struggle with issues of interoperability [5]. In its simplest form, interoperability “enables the secure exchange of health information with, and use of electronic health information from, other health information technology without special effort on the part of the user” [6]. If physicians could quickly check to see if the patient had recently received the same prescription from another physician outside the parameters of proper medication use, the practice of doctor shopping could be curtailed. Our team of researchers wondered if HIT was currently being leveraged to control for doctor shopping, if there are public health benefits to doing so, and if other researchers had already identified this problem.

An internationally focused systematic review in 2019 examined the characteristics of doctor shoppers [7]. This review found 40% of 43 studies focused on opioids, antidepressants, or psychoactive drugs, and 60% surrounded chronic disease. Only 0.5% originated from the U.S., while 25% originated from Japan. Contributing behaviors to doctor shopping were comorbidities, active substance abuse, greater distance from healthcare facility, younger age, longer disease, and poor patient satisfaction. A U.S. focused review in 2018 examined the association between prescription drug monitoring programs (PDMP) and nonfatal and fatal drug overdoses [8]. It found 47% of the 17 articles implemented PDMP only, 12% implemented program features, and 41% implemented both. Low-strength evidence existed for an association between PDMP and fatal overdoses, and program features were strongly associated a decrease in overdose deaths. An EU focused review from 2012 examined misuse of medicines and mentions doctor shopping [9]. It listed opioids analgesics, methadone, buprenorphine, and z-drugs as the chief drugs misused. It also identified the international drug control system as a control for abuse.

The United Nations’ International Narcotics Control Board predicts that misuse of prescription drugs will exceed illicit drug use soon [10]. In 01 January 2013, Americans reported misuse or abuse of prescription painkillers and 17,000 American died of overdose from painkillers [11]. Prescription drug abuse in the U.S. is second only to marijuana use across all age groups and close to 1.4 million Germans are dependent on prescription drugs. Overdose deaths involving prescription opioids in the U.S. were five times higher in 2016 than in 1999 [12].

Our research investigates doctor shopping and the HIT tools available to address it. Properly designed, HIT enables healthcare providers and patients to electronically access systems, records, or shared information in order to improve quality, safety, and effective care [6]. An example is the prescription drug monitoring program (PDMP): an electronic database in the U.S. that tracks controlled substance prescriptions in a state [13]. Another example can be seen in a health information exchange (HIE): the mobilization of health care information electronically across organizations within a region, community, or hospital system [14]. Outside the U.S., PharmaCloud is a system that enables physicians at contracted medical services providers to search patients’ medication records over the previous three months to prevent drug shopping [15]. Lastly, smart cards (NHI-IC cards) carry information about a patient’s prescribed medications received from different hospitals nationwide. This system can address the problem of duplicate medications for outpatients visiting multiple hospitals [16]. 

In addition to the HIT solutions associated with doctor shopping, some key terms for this research include multiple provider episodes (MPE), which is the obtainment of controlled substances from some minimum number of prescribers and/or pharmacies in a given period of time [17]; nonmedical use of prescription medications (NMPM) is the use of medications without a prescription from a health care provider, use in a manner other than as directed [18]; and national provider identifier (NPI) is a unique 10-digit identification number issued to healthcare providers in the United States by the Centers for Medicare and Medicaid Services [19]. Our research investigates why doctor shopping is dangerous and how health outcomes affect patient populations. Identifying doctor shopper drug shopping patterns with the use of HIT should theoretically help providers better understand and control this phenomenon. 

### 1.2. Objectives

The objective of this systematic review is to determine the prevalence of doctor shopping and how HIT can be useful in reducing it. Our research demonstrates that doctor shopping is an international concern where abuse and addiction of drugs is prevalent. The lack of an integrated health information exchange makes it difficult for providers to gain the needed information to address if an individual has already received a medication or service [14]. This review should help identify other means to control doctor shopping and their effectiveness through the examination of medical outcomes. 

## 2. Materials and Methods 

### 2.1. Protocol and Registration

This review followed the Kruse protocol published in 2019 [20]. It was reported in accordance with preferred reporting items for systematic reviews and meta-analysis (PRISMA) [21]. This review was registered with PROSPERO on 2 May 2020. In accordance with rules at PROSPERO, the registration was completed before analysis began. 

### 2.2. Eligibility Criteria

Studies in this review were eligible if some form of HIT was implemented in the control of doctor shopping, they were published in quality journal (peer reviewed), and were published in the last 10 years. Preferably, these studies also reported medical outcomes, but that was not a requirement for selection. Moreover, 10 years was chosen as the time frame because in the realm of technology, 10 years is enough time to capture current trends without confounding the results with outdated technology, and because 10 years was used in the systematic reviews referenced in our introduction section. A quality assessment of articles was made with the Johns Hopkins nursing evidence-based practice rating scale (JHNEBP) [22]. 

### 2.3. Information Sources

Reviewers queried four databases: The Cumulative Index of Nursing and Allied Health Literature (CINAHL), PubMed (MEDLINE), Web of Science, and Embase (Science Direct). Databases were filtered for the last 10 years. Database searches occurred between 1–15 February 2020. 

### 2.4. Search

Reviewers conducted a Google Scholar search using general terms about the topic. The 10 most recent articles were identified on the subject and reviewers collected the key terms from these studies to help form a Boolean search string. Using the PubMed Medical Subject Headings (MeSH), reviewers used the terms gathered from the 10 articles to examine how they were indexed and categorized. Once a Boolean search string was assembled, it was tested out several times in PubMed and customized for maximum, most effective yield. The final search string was (“doctor shopping” OR “drug shopping”) AND (“health information technology” OR “health information exchange” OR informatics). The same string was used for all four databases. Reviews were filtered out and other filters were used to help the search focus on quality articles on the subject. 

### 2.5. Study Selection

Reviewers followed the Kruse Protocol for conducting a systematic review through the use of three consensus meetings [20]. Search results from the four databases were downloaded to a common Excel spreadsheet used as a literature matrix. This spreadsheet was used throughout the process to extract data and analyze results. This piloted form had several standard data fields to collect at each stage of the process. The group leader assigned workload so all abstracts were screened by at least two reviewers against the objective statement. Reviewers made independent recommendations to keep or discard using the following codes to document their work: D = Duplicate, NJ = non-journal, P = Protocol, B = Book, R = Review, M = Model, and NG = Not Germane. During the first consensus meeting, disagreement with recommendations was discussed. A tie was broken through a third reviewer’s assessment of the article’s applicability. By the end of the meeting, a final set of articles was identified to analyze. A kappa statistic was calculated from this process [23,24]. 

### 2.6. Data Collection Process

In preparation for the second consensus meeting, the group leader assigned workload to ensure all articles were analyzed by at least two reviewers. Reviewers read through articles twice: (1) to collect participants, intervention, comparison, outcome, and study design (PICOS) data and (2) to make observations relevant to the objective. During the second consensus meeting, reviewers discussed their observations. From the collective set of observations, a thematic analysis was performed to make sense of the data [25]. This served two purposes: (1) to make sense of the observations and (2) to find synergistic effects that result in “ah ha” moments when reviewers remember reading about a theme but did not record it fully. Reviewers then carefully read through the articles a third time to make more detailed observations relevant to the themes. During the third consensus meeting, inferences were made through observed interactions between the themes. 

### 2.7. Data Items

Through the spreadsheet that served as a piloted form [20], standard data items were collected: participants, intervention (health information technology), study design, results compared to a control group (where applicable), facilitators and barriers to the use of health information technology, medical outcomes, sample size, bias within studies, effect size, country of origin, statistics used, a quality assessment from the JHNEBP [22], and general observations about the article that would help interpret the results. 

### 2.8. Risk of Bias within and across Studies

Along the process of extracting data, general observations of bias and quality were made by each reviewer. Bias, such as selection bias, was discussed in the second consensus meeting. These were important to observe because bias can limit the external validity of the results from studies. Quality assessments from the JHNEBP were also discussed in the second consensus meeting. The JHNEBP has existed since 2007. It is comprised of five levels for strength of evidence and three levels for quality of evidence. The levels under strength of evidence are as follows: level 1 is an experimental study or randomized control trial (RCT); level 2 is quasi-experimental studies; level 3 is non-experimental, qualitative, or meta-synthesis studies; level 4 is opinion of nationally recognized experts based on research evidence or consensus panels; and level 5 is opinions of experts that is not based on research evidence. The levels under quality of evidence are as follows: A (high), B (good), or C (low quality or major flaws). Under each of these levels, specifics are defined for research, summative reviews, organizational, and expert opinion; e.g., research in level A must have consistent results with sufficient sample size, adequate control, and definitive conclusions; research in level B must have reasonably consistent results, sufficient sample size, some control, and definitive conclusions; research at level C has little evidence with inconsistent results, insufficient sample size, and conclusions that cannot be drawn from the data. Articles with a strength of evidence rating below Level 4 will be screened out. Quality of evidence below level B are highly suspect and must have full consensus of the group to be kept for analysis.

### 2.9. Summary Measures

The review analyzed studies with qualitative, quantitative, and mixed methods, so the summary measures sought were not consistent. The preferred summary statistic would be the risk ratio, but descriptive statistics, means’ comparisons (student-t) are also sufficient. Summary statistics were also discussed at the second consensus meeting. 

### 2.10. Additional Analysis

At the second consensus meeting, a thematic analysis was performed to group observations into themes. These themes were measured across all articles analyzed and reported in summary statistics in a series of affinity matrices. The thematic analysis summarized themes for facilitators, barriers, and medical outcomes. These are reported in affinity matrices in the Results section.

## 3. Results

### 3.1. Study Selection

The study selection process performed is illustrated in Figure 1. A kappa statistic was calculated after the first consensus meeting (*k* = 0.95), which indicates near perfect agreement [23,24]. After screening, removing duplicates and assessing for eligibility, the 48 articles chosen for analysis came from CINAHL (1, 2%), Web of Science (1, 2%), PubMed (28, 58%), and Science Direct (18, 38%).

### 3.2. Study Characteristics

Using the piloted form, reviewers collected several standard items used for summary, such as PICOS. A PICOS table is provided in Table 1. Additional items were collected for analysis, such as forms of assistive technology interventions, facilitators, and barriers to the use of assistive technologies, and the medical outcomes observed from those older adults using assistive technology solutions. These are presented in Table 2. Table 1 and Table 2 lists articles in reverse chronological order: 2020 (1) [26], 2019 (7) [27,28,29,30,31,32,33], 2018 (4) [34,35,36,37], 2017 (3) [38,39,40], 2016 (7) [17,41,42,43,44,45,46], 2015 (4) [15,47,48,49], 2014 (3) [11,14,50], 2013 (2) [51,52], 2012 (4) [53,54,55,56], 2011 (6) [13,16,57,58,59,60], 2010 (1) [61].

### 3.3. Risk of Bias within Studies

Reviewers recorded observations of bias at the study level. The most common form of bias was convenience samples taken from one country only. This was common to every article in the review. This is logical since countries struggle enough with domestic interoperability. International interoperability may be too much to ask for in the near term. There was one instance of selection bias [40]. These examples of bias limit the external validity of the results.

### 3.4. Results of Individual Studies

Reviewers collected their observations of intervention and medical outcomes during the analysis phase. The narrative analysis of their observations identified themes. A summary of these themes is listed in Table 2. Repetition in the frame of a theme is due to multiple observations from the same article for that theme. For instance, the theme increased talking comprised observations of “increased utterances” and “increased sustained conversations.” A translation from observations to themes for interventions, medical outcomes, facilitators, and barriers are listed in Appendix A. Additional data items extracted are displayed in Appendix B: sample size, bias, country of origin, statistics, and quality assessments.

### 3.5. Synthesis of Results

This subsection addresses meta-analyses. This manuscript is a systematic review. This subsection will be deleted after the review process. It is included to reassure reviewers that we followed the PRISMA checklist.

### 3.6. Risk of Bias across Studies

Table 3 summarizes the quality indicators identified by the JHNEBP tool [22]. The most prevalent assessment in the strength of evidence (panel a) was level III, followed by IV, and II. For quality of evidence (panel b), the most frequently assessed level was level A, followed by B and C. It is certainly preferable for the strength of evidence be closer to level I, but that result did not materialize from the screening and selection process. This limitation will be addressed later.

### 3.7. Additional Analysis

#### 3.7.1. Interventions of HIT

The results of consensus meeting three identified seven intervention themes that corresponded with utilizing HIT to control for doctor shopping. These are listed in Table 4. In the interest of brevity, only the top 90% are described. The intervention PDMP (state-run prescription drug monitoring program or interviews/surveys about PDMP) appeared in 15/42 articles (36%) [13,26,28,31,38,39,41,43,49,50,51,54,55,56,60]. These articles all originated from the United States because that is the only country that uses PDMP. The intervention national health system database (European, Scotland, French, Japan, Finland) appeared in 7/42 articles (17%) [17,36,37,42,44,53,61]. The intervention computer model (e.g., clustering models, PageRank, social network analysis) appeared in 5/42 articles (17%) [29,30,32,46,59]. Three of these articles originated from the United States, while the others were from Japan and France. The intervention combination (PDMP combined with worker’s compensation claims, PDMP and Lock-in programs, PDMP education efforts and pharmacist panels, PDMP and PBM, Computerized Provider Order Entry (CPOE) and NHI-IC cards) appeared in 5/42 articles (17%) [11,16,27,34,54]. Four of these articles originated in the United States, while the other was from Taiwan. The intervention other (pharmacological database, medication record sharing program, PharmaCloud, ACOS and DAWN, and CEIP) appeared in 5/42 articles (17%) [15,33,35,52,57]. Two of these articles originated from the United States, two from Taiwan, and one from France.

#### 3.7.2. Facilitators of HIT

The results of consensus meeting three identified five themes and two individual observations that corresponded with facilitators of HIT to control for doctor shopping. These are listed in Table 5, which contains Table 5, Table 6 and Table 7 (facilitators, barriers, and medical outcomes). In the interest of brevity, only the first 75% will be listed (other than not reported). The facilitator theme was government support (state supports prescription monitoring or prescription monitoring programs) occurred in 31/52 occurrences (60%) [13,14,17,27,28,31,33,35,36,37,39,40,41,42,43,44,45,47,48,49,52,53,54,55,56,57,58,59,60,61]. In total, 17 of these articles originated from the United States, seven from France, three from Taiwan, while the others were from original originations in Finland and Scotland. The theme prescriber support (physician, nurse practitioner) occurred 6/52 times (12%) [28,31,45,55,56]. Four of these articles originated from the United States while the other was from Taiwan.

#### 3.7.3. Barriers of HIT

The results of the consensus meeting identified three themes and six individual observations that corresponded with barriers of HIT to control for doctor shopping. These are listed in Table 6. In the interest of brevity, only the themes will be discussed (other than not reported). The theme of data sharing (PDMP do not all share across state lines) occurred in 5/52 occurrences (10%) [11,27,39,45,51]. These articles all originated from the United States. The theme access not mandatory (access to PDMP or PharmaCloud not mandatory and the intervention is not widely implemented because of this fact) occurred in 4/52 occurrences (8%) [15,28,34,49]. Three of these articles originated from the United States while the other originated from Taiwan. The theme participation was not mandatory (not all states participate in health information exchanges, not all facilities participate in PharmaCloud) occurred in 3/52 occurrences (6%) [15,16,58]. Two of these articles originated in Taiwan, while the other was from the United States.

#### 3.7.4. Medical Outcomes Commensurate with HIT as Intervention

The results of consensus meeting three identified two themes and four individual observations that corresponded with medical outcomes commensurate with HIT to control for doctor shopping. These are listed in Table 7. In the interest of brevity, only the themes will be listed (other than not reported). The theme reduced [doctor] shopping (decreased shopping and diversion) occurred in 3/42 articles (7%) [50,51,55]. These originated from the United States. The theme reduced mortality (decline in overdose mortality) occurred in 2/42 articles (5%) [34,40]. These originated from the United States.

### 3.8. Interactions between Observations

The intervention of the national health system database resulted in seven instances of government support [17,36,37,42,44,53,61]. The intervention of the computer model resulted in five instances of simple to implement [29,30,32,46,59]. The intervention PDMP resulted in two instances of access not mandatory [28,49] and two instances of data sharing [39,51].

## 4. Discussion

### 4.1. Summary of Evidence

Through the analysis of 42 articles, this review identified seven interventions, five facilitator themes with two individual observations, three barrier themes with six individual observations, and two medical outcome themes with four individual observations. Most of the articles analyzed were non-experimental (28/42, 67%) but of high quality (25/42, 60%). The facilitator themes mentioned most frequently were government support (31/52, 60%) [13,14,17,27,28,31,33,35,36,37,39,40,41,42,43,44,45,47,48,49,52,53,54,55,56,57,58,59,60,61] and prescriber support (5/52, 12%) [28,31,45,55,56]. The barrier themes mentioned most frequently were data sharing (5/48, 10%) [11,27,39,45,51] and access not mandatory (4/48, 8%) [15,28,34,49]. The medical outcome themes mentioned most frequently were reduced shopping (3/42, 7%) [50,51,55] and reduced mortality (2/42, 5%) [34,40]. A large majority originated from the United States (26/42, 62%), but also from France (7/42, 17%), Taiwan (5/42, 12%), Japan (2/42, 5%), and one each from Finland and Scotland.

The high level of government and prescriber support was reassuring to observe because it indicates that government health agencies recognize the problem of doctor shopping and are attempting to address it through regulation. It also indicates that governments and prescribers are communicating about the problem and collaborating for solutions. This result was expected because they had been observed in other research [7,8,9].

One interesting dichotomy was observed. In medical outcomes, 2/42 articles mentioned a reduction in overdose mortality commensurate with the use of HIT measures to control for doctor shopping [34,40], while one listed no decline in mortality [54]. A key point made by one article is that the control programs like PDMP do nothing to treat the disease of addiction driving the behavior [34]. This author mentioned that unless control programs institute treatment [26], doctor shoppers will find other means of finding their illicit drugs to misuse.

Policy makers should examine the barriers listed in this review. The barrier of data access predominantly spoke of the United States with state-run PDMPs. All but three states share data, but Florida, Georgia, and Nebraska do not allow data sharing [49]. Without nationwide data sharing agreements, shoppers just travel to the next state to do their shopping. Shoppers are documented to travel an average of 199.5 miles to shop, while non-shoppers do not travel at all [51]. Another key barrier is that participation is not mandatory. This occurred in the he U.S. with PDMP and Taiwan with PharmaCloud. If participation is not mandatory, universal enforcement cannot occur.

### 4.2. Limitations

The researchers analyzed articles from 2010 to 2020. The reviewers hoped to find higher strength studies from which to extract data and summarize results. However, there is a paucity of high-strength studies in this area. Future researchers should consider this high-need area to explore.

This review limited its search to four databases: PubMed, CINAHL, Science Direct, and Web of Science. It did not include broader sources such as Google Scholar. Even though 42 articles were analyzed, it is possible other sources may have yielded higher strength articles to include in the analysis. Only 67% were high quality articles, while the others (17) were of good or low quality. It is important to base conclusions on high quality articles, but often we are relegated to what is already published.

A team of reviewers determined the articles to be included in the study. This was also done to mitigate the risk of selection bias. The risk of this practice, however, is that the team may have differed in their selection process. To mitigate this risk, researchers held consensus meetings and identified the research objective, and had multiple reviews for each article. The limitation is that there may not have been enough consensus meetings. The kappa statistic shows the consensus meetings were effective, yet a stronger level of agreement is possible.

## 5. Conclusions

This research supports the use of HIT as a control for doctor shopping. Computer models are simple to implement and monitoring systems exist to help prescribers and dispensers’ control for doctor shopping behavior. Greater interaction, whether voluntary or mandatory, yields greater success as the behavior is identified. After identifying the behavior, treatment and help should follow. Otherwise, the patient will find other outlets for the behavior. Government intervention is evident in the research, and this enabler should be exploited. Robust data sharing and both participation and consultation of PDMP should be a standard in the industry to decrease doctor shopping and improve mortality.

## Figures and Tables

**Figure 1 healthcare-08-00306-f001:**
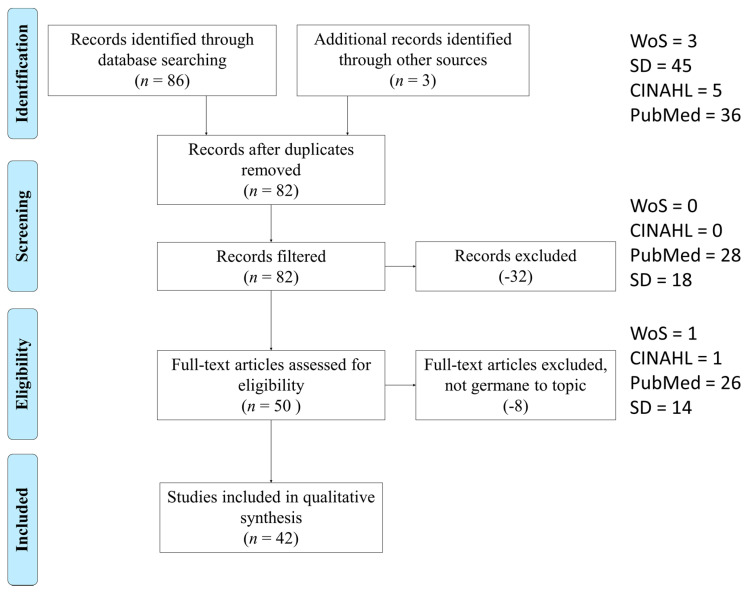
Preferred reporting items for systematic reviews and meta-analysis (PRISMA) figure that demonstrates the study selection process.

**Table 1 healthcare-08-00306-t001:** PICOS for all articles analyzed.

Authors	Participants	Intervention	Results (Compared to Control Group)	Medical Outcomes Reported	Study Design
Pett RG et al. [26]	Pharmacists	PDMP	Frequent PDMP users were more likely to recommend naloxone.	Naloxone prescribed to shoppers	Explanatory, sequential 2-phase mixed-methods
Durand et al. [27]	Injured workers identified in worker’s compensation records	PDMP and worker’s compensation claims	No control group. Injured workers have a high prevalence of opioid use after injury, but prescribing patterns generally follow state guidelines.	Not reported	Retrospective cohort
Freeman et al. [28]	Pharmacists and providers	Interviews about PDMP	No control group. Both PCPs and pharmacists reported PDMPs are key tools to aid prescribing and dispensing.	Not reported	Qualitative
Nagarajan and Talbert [29]	Prescriber-prescriber networks	Computer model	Outliers were clearly identified in the model, which can help identify those prescribers contributing to the opioid epidemic.	Not reported	Computer model to detect prescriber outliers
Perry et al. [30]	Patients who submitted claims to commercial database in the Appalachian region of the U.S., 58.7% female	Computer model PageRank	Model clearly differentiates aberrant behavior identifying drug shoppers for both opioids and morphine milligram equivalents (MME).	Not reported	Computer model to detect prescriber outliers
Soffin et al. [31]	n/a	PDMP	n/a	Accessing real-time information about patients’ prescription opioid status using PDMP reduces opioid quantities prescribed	Opinion
Stopka et al. [32]	adults in Massachusetts prescribed opioids	Computer model	Hotspots were identified.	Not reported	Spatial epidemiological study
Wang et al. [33]	Patients in pharmacological database in Florida	Pharmacological database	Recipients of opioids, benzodiazepines, and carisoprodol in 2017 compared with 2012 were younger, more likely to be female, and geographically-localized.	Not reported	Retrospective, observational
Butler et al. [34]	n/a	PDMP, Lock-in programs	No control group. The lock-in program decreased doctor shopping.	Broader range of drug schedules and an increases frequency of updating the PDMP results in lower opioid-related mortality	Opinion
Lin et al. [35]	Patients from health insurance claim data	Medication record sharing program	The medication duplication rate was reduced 7.76 percentile, average medication overlap periods shortened 4.36 days.	Not reported	Retrospective pre-post test design
Ponte et al. [36]	Beneficiaries of the France public health system	SNIIRAM database	The strong opioid analgesics have the highest DSI (2.79%) versus 2.06% for BZD hypnotics. Flunitrazepam ranked first according to its DSI (13.2%), followed by morphine (4%), and zolpidem (2.2%).	Not reported	Retrospective, observational
Torrance et al. [37]	Beneficiaries of the Scotland National Health System (NHS)	NHS and Generation Scotland databases combined to identify trends	The number of strong opioid prescriptions more than doubled between 2003–2012. Patients in the most deprived areas were more likely to receive a strong opioid.	Not reported	Descriptive analysis
Ali et al. [38]	Respondents to the national survey of drug use and health, aged 12 and older, 48% male	Survey instrument to assess effectiveness of the PDMP	No control group. PDMP was effective at controlling for doctor shopping for opiate pain killers.	10–20 fewer days of NMPR use	Qualitative
Rutkow et al. [39]	Prescribers in four states	PDMP	Prescribers need to work with law enforcement, law-enforcement need to share data with each other, data sharing between states needs to occur.	Not reported	Qualitative
Simeone [40]	Prescriptions	PDMP, education efforts, pharmacy panels that span the country	No control group. The number of prescriptions diverted fell from 4.3 million in 2008 to 3.37 million in 2012.	Decline in overdose mortality	Retrospective, observational
Chenaf et al. [17]	Adult patients with chronic non-cancer pain (CNCP)	French national health system	No control group. Shopping very low in drugs for these conditions.	Not reported	Retrospective cohort
Delorme et al. [42]	Patients treated by opioid substitution treatment over 8 years	French national health system	No control group. Shopping behavior was only found in high dosage buprenorphine patients, but still very low.	Not reported	Retrospective cohort
Kea et al. [43]	Emergency Department (ED) discharges	PDMP	Doctor shopping was not detected in ED survey.	Not reported	Qualitative

**Table 2 healthcare-08-00306-t002:** Summary of analysis.

Authors	Intervention Theme	Outcome Theme	Facilitator Theme	Barrier Theme
Pett et al.	PDMP	Shoppers prescribed treatment	Pharmacists support	Not reported
Durand et al.	Combination	Not reported	Government support	Data sharing
Freeman et al.	PDMP	Not reported	Government support	Access not mandatory
			Pharmacists support	
			Prescriber support	
Nagarajan and Talbert	Computer model	Not reported	Simple to implement	Not reported
Perry et al.	Computer model	Not reported	Not reported	Not reported
Soffin et al.	PDMP	Reduced opioids prescribed	Government support	Not reported
			Prescriber support	
Stopka et al.	Computer model	Not reported	Simple to implement	Not reported
Wang et al.	Other	Not reported	Government support	Not reported
Butler et al.	Combination	Reduced mortality	Must use law	Access not mandatory
				Does not treat addiction
Lin et al.	Other	Not reported	Government support	Not reported
Ponte et al.	National health system database (DB)	Not reported	Government support	Not reported
Torrance et al.	National health system DB	Not reported	Government support	Not reported
Ali et al.	PDMP	Fewer days of use	Not reported	Not reported
Rutkow et al.	PDMP	Not reported	Government support	Data sharing
Simeone	Combination	Reduced mortality	Government support	Easily thwarted
			Pharmacists support	
Chenaf et al.	National health system DB	Not reported	Government support	Not reported
Delorme et al.	National health system DB	Not reported	Government support	Not reported
Kea et al.	PDMP	Not reported	Government support	Not reported
National Council of State Boards of Nursing	PDMP	Not reported	Government support	Not reported
Okumura et al.	National health system DB	Not reported	Government support	Not reported
Ong et al.	Health insurance claims	Not reported	Government support	Data sharing
			Prescriber support	
Takahashi et al.	Computer model	Not reported	Simple to implement	Not reported
Huang et al.	Other	Not reported	Cost savings	Participation not mandatory
			Government support	Access not mandatory
Lin et al.	Health insurance claims	Not reported	Government support	Not reported
Lu et al.	Health insurance claims	Not reported	Government support	Not reported
Webster and Grabois	PDMP	Not reported	Government support	Access not mandatory
Han et al.	PDMP	Reduced shopping	Not reported	Not reported
Hypponen et al.	Health information exchange	Not reported	Government support	Not reported
			Cost savings	
			Increased efficiency	
Shepherd	Combination	Not reported	Not reported	Inadequate data collection
				Ineffective data use
				Data sharing
				Constraints on enforcement
				Cash-only not captured
Cepeda et al.	PDMP	Reduced shopping	Not reported	Data sharing
Modarai et al.	Other	Not reported	Government support	Not reported
Rouby et al.	National health system DB	Not reported	Government support	Not reported
Simoni-Wastila and Qian	PDMP	No decline in mortality	Government support	Not reported
Worley et al.	PDMP	Reduced shopping	Government support	Not reported
			Prescriber support	
Worley et al.	PDMP	Not reported	Government support	Not reported
			Prescriber support	
Fass and Hardigan	PDMP	Not reported	Government support	Not reported
Frauger et al.	Other	Not reported	Government support	Not reported
Hincapie et al.	Health information exchange	Not reported	Government support	Participation not mandatory
Hsu et al.	Combination	Not reported	Prescriber support	Participation not mandatory
Pauly et al.	Computer model	Not reported	Government support	Not reported
Wilsey et al.	PDMP	Not reported	Government support	Not reported
Pradel et al.	National health system DB	Not reported	Government support	Not reported

**Table 3 healthcare-08-00306-t003:** Summary of quality assessments.

Strength of Evidence	Frequency	Quality of Evidence	Frequency
III (Non-experimental, qualitative)	28 (67%)	A (High quality)	25 (60%)
IV (Opinion)	10 (24%)	B (Good quality)	11 (26%)
II (Quasi-experimental)	4 (10%)	C (Low quality or major flaws)	6 (14%)
I (Experimental study or RCT)	0 (0%)		
(a)		(b)	

**Table 4 healthcare-08-00306-t004:** Affinity matrix of Health Information Technology (HIT) interventions to control for doctor shopping.

Interventions	References	Occurrences (*n* = 42)	Frequency
PDMP	[13,26,28,31,38,39,41,43,49,50,51,54,55,56,60]	15	36%
National health system DB	[17,36,37,42,44,53,61]	7	17%
Computer model	[29,30,32,46,59]	5	12%
Combination	[11,16,27,34,54]	5	12%
Other	[15,33,35,52,57]	5	12%
Health insurance claims	[45,47,48]	3	7%
Health information exchange	[14,58]	2	5%

**Table 5 healthcare-08-00306-t005:** Affinity matrix of facilitators of Health Information Technology (HIT)to control for doctor shopping.

Facilitators	References	Occurrences (*n* = 52)	Frequency
Government support	[13,14,17,27,28,31,33,35,36,37,39,40,41,42,43,44,45,47,48,49,52,53,54,55,56,57,58,59,60,61]	31	60%
Prescriber support	[28,31,45,55,56]	6	12%
Not reported	[11,30,38,50,51]	5	10%
Simple to implement	[29,30,32,46]	4	6%
Pharmacists support	[26,28,40]	3	6%
Cost savings	[14,15]	2	4%
Must use law	[34]	1	2%
Increased efficiency	[14]	1	2%

**Table 6 healthcare-08-00306-t006:** Affinity matrix of barriers of Health Information Technology (HIT)to control for drug shopping.

Barriers	References	Occurrences (*n* = 48)	Frequency
Not reported	[13,14,17,26,29,30,31,32,33,35,36,37,38,41,42,43,44,46,47,48,50,52,54,55,56,57,59,60,61]	30	63%
Data sharing	[11,27,39,45,51]	5	10%
Access not mandatory	[15,28,34,49]	4	8%
Participation not mandatory	[15,16,58]	3	6%
Easily thwarted	[40]	1	2%
Inadequate data collection	[11]	1	2%
Ineffective data use	[11]	1	2%
Constraints on enforcement	[11]	1	2%
Cash-only not captured	[11]	1	2%
Does not treat addiction	[34]	1	2%

**Table 7 healthcare-08-00306-t007:** Affinity matrix of medical outcomes commensurate with Health Information Technology (HIT) as intervention.

Medical Outcomes	References	Occurrences (*n* = 42)	Frequency
Not reported	[11,13,14,15,16,17,27,28,29,30,32,33,35,36,37,39,41,42,43,44,45,47,48,49,52,53,56,57,58,59,60,61]	33	79%
Reduced shopping	[50,51,55]	3	7%
Reduced mortality	[34,40]	2	5%
Shoppers prescribed treatment	[26]	1	2%
Fewer days of use	[38]	1	2%
No decline in mortality	[54]	1	2%
Reduced opioids prescribed	[31]	1	2%

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
