# Peer review of "Health Information Technology and Doctor Shopping: A Systematic Review"

_healthcare, 2020, doi:10.3390/healthcare8030306_

Round 1
Reviewer 1 Report
Review of Healthcare 894275
Thank you for the opportunity to peer review this systematic review. This body of work was extensive, and I applaud those involved. With this said, the manuscript has a lot of issues within the paper. The first and foremost concern is the lack of accurate citation and referencing of the Johns Hopkins Nursing Evidence-Based Practice Model and Guidelines. Please address all concerns.
Abstract is well written. The conclusion is severely lacking in comparison to what is concluded in the abstract or the summary.
Page 1-2; last and first paragraph: This first paragraph of the introduction section is very hard to read. The first statement about citizen empowerment leads the opening dialog and is lost throughout. The second sentence is as misleading. Both need to be placed elsewhere or the point made more clearly. The broad statements leading to a very fine point of HIT leveraging is disconnected. Please redo the whole paragraph so the point is broad to understand the scope of problem and then fine tuned to what you are trying to investigate.
All first six sentences are referencing other works, but do not connect well together. Reorganizing your first paragraph is needed to understand the point of the study. It is a poor entry into your rationale.
Line 37-38: “The simple answer lies …….”. Stating this makes the reader think that you have figured out the problem, so why write a systematic review? This broad statement of fact should be deleted or presented as a possible problem not a statement of fact.
Page 2, line 57: first sentence needs to be explained why the authors think this review is self-evident, this is not obvious to reader. Connect to the evidence in the same paragraph.
Page 2, line 58: misuse of prescription drugs will exceed illicit drug use soon. Quantify soon – what does this mean? What did the author say about timeline?
Page 2, lines 63-72. Again, the sentences are not connected. In the first sentence, the subject is about HIT enabling providers to connect with electronic records. Then I can only assume the next few sentences are examples of this. But the way the first is written it does not lead me to connect the next examples. Rewrite this please.
Page 2, lines 72-77. Other key terms for this research….. What were the first terms of the research? It is not clear.
Page 2, lines 77-78. Our research investigates…. This should be the beginning sentence of this paragraph. It would help to explain why you are researching the problem. Then talk about HIT, etc. Ending this paragraph with the last statement would be appropriate if the rest of paragraph were reorganized and rewritten as suggested.
Page 3, lines 99-100. The listed reference Johns Hopkins Nursing …. (22) is not an accurate reference for the EBP rating scale. The scale, model or guidelines are not within this reference.
Page 3, lines 104-105: last sentence is not appropriate for ‘information sources’. Rewrite please. Delete “we expected to…”.
Page 3, lines 106-107: on page 14, lines 305-306 it says the authors did not include broader sources such as Google Scholar. Here is says you did. Please clarify.
Page 4, lines 148-162. ‘The JHNEBP…. This section is quoted directly from a source not listed. This is a major omission of this systematic review.
Page 11, line 214. Again, the reference listed is not the reference for the JHNEBP tool.
Page 11, lines 214-220. The levels of evidence and strength of evidence are used without reference. Table 3 does not include a synthesis of findings for each level. It you are using JHNEBP, it is highly suggested that you use the appropriate Synthesis and Recommendations Tool by JHNEBP.
Page 14, lines 280-281. ‘Most of the articles analyzed were non-experimental but of high quality. This statement tells me that only 67% were non-experimental, a little over half, which means 33 % were consensus/position statement articles (Level 4) or literature reviews without evidence of quality of strength, expert opinions, community standard, consumer experience, etc. (Level 5). From this statement we do not know which ones were of high quality (60%) or good or low quality. It is suggested creating a Synthesis Process and Recommendations Tool for the articles you chose to use in this systematic review. This is valuable as it provides strength and evidence to your systematic review.
Page 11 and Page 14: The limitation stated would be addressed later in “limitations section” is limited to the paucity of high strength leveled studies. The concern that is not mentioned is the paucity of high ‘quality articles’, 26% and 6% or 17 articles were of good or low quality. Please address this as well. It is very important to have high level quality in an article as well as the strength.
The conclusion is very minimal, limited to what could be done to alleviate the problem of doctor shopping. It does not include any conclusion from this extensive systematic review. It is missing the elements of barriers, interventions, etc. that was included in the abstract.
Reformatting of references: 10, 12, 15, 22, 31, 36, 42, 47, 55 (looks like they were cut and pasted from another document).
Author Response
- The first and foremost concern is the lack of accurate citation and referencing of the Johns Hopkins Nursing Evidence-Based Practice Model and Guidelines. Please address all concerns.
- We changed the reference to be the 2005 document. However, access to this document is highly restricted
- The conclusion is severely lacking in comparison to what is concluded in the abstract or the summary.
- I think I understand your point. We spent very little time on addiction, and that is not the focus of the systematic review. I changed the conclusion to focus more on data sharing.
- This first paragraph of the introduction section is very hard to read. The first statement about citizen empowerment leads the opening dialog and is lost throughout. The second sentence is as misleading. Both need to be placed elsewhere or the point made more clearly.
- I removed the sentences. I agree they caused confusion and most likely do not improve the quality of the manuscript.
- (more on the first paragraph) The broad statements leading to a very fine point of HIT leveraging is disconnected. Please redo the whole paragraph so the point is broad to understand the scope of problem and then fine tuned to what you are trying to investigate.
- Removing the first two sentences already greatly improved this paragraph. I also clarified some of the language in the paragraph.
- All first six sentences are referencing other works, but do not connect well together. Reorganizing your first paragraph is needed to understand the point of the study. It is a poor entry into your rationale.
- There is nothing actionable in this bullet that is not already stated in the previous one.
- “The simple answer lies …….”. Stating this makes the reader think that you have figured out the problem, so why write a systematic review? This broad statement of fact should be deleted or presented as a possible problem not a statement of fact.
- We softened this statement. We did not intend to present a solution in the introduction.
- Page 2, line 57: first sentence needs to be explained why the authors think this review is self-evident, this is not obvious to reader. Connect to the evidence in the same paragraph.
- We removed the sentence.
- Page 2, line 58: misuse of prescription drugs will exceed illicit drug use soon. Quantify soon – what does this mean? What did the author say about timeline?
- "Soon" is the exact word used by the UNINCB. I agree that the language is indistinct, but there is not science to define a timeline, and we did not define one.
- Page 2, lines 63-72. Again, the sentences are not connected. In the first sentence, the subject is about HIT enabling providers to connect with electronic records. Then I can only assume the next few sentences are examples of this. But the way the first is written it does not lead me to connect the next examples. Rewrite this please.
- Rewritten, as requested.
- Page 2, lines 72-77. Other key terms for this research….. What were the first terms of the research? It is not clear.
- Clarified, as requested.
- Page 2, lines 77-78. Our research investigates…. This should be the beginning sentence of this paragraph. It would help to explain why you are researching the problem. Then talk about HIT, etc. Ending this paragraph with the last statement would be appropriate if the rest of paragraph were reorganized and rewritten as suggested.
- Reworded, as requested.
- Page 3, lines 99-100. The listed reference Johns Hopkins Nursing …. (22) is not an accurate reference for the EBP rating scale. The scale, model or guidelines are not within this reference.
- This is addressed in in bullet number 1.
- Page 3, lines 104-105: last sentence is not appropriate for ‘information sources’. Rewrite please. Delete “we expected to…”.
- We deleted the sentence.
- Page 3, lines 106-107: on page 14, lines 305-306 it says the authors did not include broader sources such as Google Scholar. Here is says you did. Please clarify.
- Google Scholar is not a research database. It was not used for a source of research (information sources). It was used, as stated, to discover general terms associated with the topic of doctor shopping. This is further explained in the published protocol.
- Page 4, lines 148-162. ‘The JHNEBP…. This section is quoted directly from a source not listed. This is a major omission of this systematic review.
- This is a simple sentence that is a statement of fact used in a manuscript only accepted for publication late last week (that I wrote). To appease your objection, we made a small change in the wording.
- Page 11, line 214. Again, the reference listed is not the reference for the JHNEBP tool.
- This is addressed in bullet 1.
- Page 11, lines 214-220. The levels of evidence and strength of evidence are used without reference. Table 3 does not include a synthesis of findings for each level. It you are using JHNEBP, it is highly suggested that you use the appropriate Synthesis and Recommendations Tool by JHNEBP.
- I disagree with your premise. The JHNEBP was designed to determine if a change in practice is needed in the nursing field. We used the JHNEBP as a quality assessment of articles analyzed. We are not intending to translate our findings to the nursing practice. We provided a reference (22) that points the reader to the source. That is sufficient for our use of the tool.
- Page 14, lines 280-281. ‘Most of the articles analyzed were non-experimental but of high quality. This statement tells me that only 67% were non-experimental, a little over half, which means 33 % were consensus/position statement articles (Level 4) or literature reviews without evidence of quality of strength, expert opinions, community standard, consumer experience, etc. (Level 5). From this statement we do not know which ones were of high quality (60%) or good or low quality. It is suggested creating a Synthesis Process and Recommendations Tool for the articles you chose to use in this systematic review. This is valuable as it provides strength and evidence to your systematic review.
- That is not correct, which is illustrated by Table 3. Only 24% were opinion: 10% quasi-experimental. Zero were literature reviews (stated). We used zero level 5 articles (also stated). To enable readers to determine strength and quality, we included it in Appendix B. We do not agree it is necessary to create a Syntheses Process and Recommendations Tool. That that is not the purpose of this research.
- Page 11 and Page 14: The limitation stated would be addressed later in “limitations section” is limited to the paucity of high strength leveled studies. The concern that is not mentioned is the paucity of high ‘quality articles’, 26% and 6% or 17 articles were of good or low quality. Please address this as well. It is very important to have high level quality in an article as well as the strength.
- This is now addressed.
-
The conclusion is very minimal, limited to what could be done to alleviate the problem of doctor shopping. It does not include any conclusion from this extensive systematic review. It is missing the elements of barriers, interventions, etc. that was included in the abstract.
- The enabler of Governement Intervention, the barriers of Data sharing, access not mandatory, and participation not mandatory, and the medical outcomes of decreased doctor shopping and improved mortality are now addressed in the conclusion.
-
Reformatting of references: 10, 12, 15, 22, 31, 36, 42, 47, 55 (looks like they were cut and pasted from another document).
- They were not. A bibliographic manager was used. I fixed the code in reference 10. Reference 12 is an Internet citation. I do not see anything wrong with the rest.
Reviewer 2 Report
Thank you. A very interesting and well written manuscript.
Please see some specific comments below:
Lines 41-43: If physicians could quickly check to see if the patient had recently received the same prescription from another physician outside the 43 parameters of proper medication use, they could curtail this practice.
Lines 139 and 269: citation is out of place (preceded by a comma or a full stop). Please review.
Tables 1 & 2 can be improved by drawing a line to separate info from different articles.
The discussion needs to elaborate on findings. Were they expected? Why? This section would also benefit from a paragraph on implications in practice (given the benefits and barriers) and authors’ recommendations.
Please review spaces across the manuscript (additional or omission of space e.g. line 274
Please review references as appropriate.
Author Response
- Lines 41-43: If physicians could quickly check to see if the patient had recently received the same prescription from another physician outside the 43 parameters of proper medication use, they could curtail this practice.
- I do not see a recommendation in this statement. We did modify this sentence, however, based on feedback from another reviewer.
- Lines 139 and 269: citation is out of place (preceded by a comma or a full stop). Please review.
- Corrected. Thank you for seeing these.
- Tables 1 & 2 can be improved by drawing a line to separate info from different articles.
- I inserted lines between articles.
- The discussion needs to elaborate on findings. Were they expected? Why? This section would also benefit from a paragraph on implications in practice (given the benefits and barriers) and authors’ recommendations.
- We added a paragraph to elaborate on the findings. A paragraph of recommendations was already included, "Policy makers should . . . "
- Please review spaces across the manuscript (additional or omission of space e.g. line 274
- Line spacing was reviewed. We attempted to use two spaces after end-of-sentence punctuation.
- Please review references as appropriate.
- References reviewed. We removed the computer code from reference 10 that was most likely an artifact of using a bibliographic management system.
Round 2
Reviewer 1 Report
Dear Authors and Editors,
I have reviewed this manuscript again, and after the important changes, I find this systematic review thorough and adequately described. Thank you for making the changes as I believe now it is publishable.